# A Novel Wall-Associated Kinase TaWAK-5D600 Positively Participates in Defense against Sharp Eyespot and *Fusarium* Crown Rot in Wheat

**DOI:** 10.3390/ijms24055060

**Published:** 2023-03-06

**Authors:** Haijun Qi, Xiuliang Zhu, Wenbiao Shen, Zengyan Zhang

**Affiliations:** 1College of Life Science, Nanjing Agricultural University, Nanjing 210095, China; 2Key Laboratory of Biology and Genetic Improvement of Triticeae Crops, Ministry of Agriculture and Rural Affairs of the People’s Republic China, The National Key Facility for Crop Gene Resources and Genetic Improvement, Institute of Crop Sciences, Chinese Academy of Agricultural Sciences, Beijing 100081, China

**Keywords:** *Fusarium pseudograminearum*, *Rhizoctonia cerealis*, sharp eyespot, wall-associated receptor-like kinase, wheat (*Triticum aestivum*)

## Abstract

Sharp eyespot and *Fusarium* crown rot, mainly caused by soil-borne fungi *Rhizoctonia cerealis* and *Fusarium pseudograminearum*, are destructive diseases of major cereal crops including wheat (*Triticum aestivum*). However, the mechanisms underlying wheat-resistant responses to the two pathogens are largely elusive. In this study, we performed a genome-wide analysis of wall-associated kinase (WAK) family in wheat. As a result, a total of 140 *TaWAK* (not *TaWAKL*) candidate genes were identified from the wheat genome, each of which contains an N-terminal signal peptide, a galacturonan binding domain, an EGF-like domain, a calcium binding EGF domain (EGF-Ca), a transmembrane domain, and an intracellular Serine/Threonine protein kinase domain. By analyzing the RNA-sequencing data of wheat inoculated with *R. cerealis* and *F. pseudograminearum*, we found that transcript abundance of *TaWAK-5D600* (*TraesCS5D02G268600*) on chromosome 5D was significantly upregulated, and that its upregulated transcript levels in response to both pathogens were higher compared with other *TaWAK* genes. Importantly, knock-down of *TaWAK-5D600* transcript impaired wheat resistance against the fungal pathogens *R. cerealis* and *F. pseudograminearum*, and significantly repressed expression of defense-related genes in wheat, *TaSERK1*, *TaMPK3*, *TaPR1*, *TaChitinase3*, and *TaChitinase4*. Thus, this study proposes *TaWAK-5D600* as a promising gene for improving wheat broad resistance to sharp eyespot and *Fusarium* crown rot (FCR) in wheat.

## 1. Introduction

As one of the most important staple cereal crops, bread wheat (*Triticum aestivum*) provides about a fifth of the total calories consumed around the world [1]. However, wheat grain production is threatened by various soil-borne viruses and fungal pathogens, such as the fungi *Rhizoctonia cerealis* and *Fusarium pseudograminearum* [2,3,4,5]. These two pathogens often cause stem necrosis and plant wilt of wheat, leading to severe production reduction and grain losses. Planting and application of resistant wheat cultivars are the most effective methods to control these soil-borne pathogens [6,7,8]. To improve wheat resistance against fungal diseases, it is vital to identify resistance or key immune-regulated genes in wheat.

Wall-associated kinase (WAK) proteins belong to a unique plant subfamily of receptor-like kinases (RLKs) and are likely linked to the cell wall by the galacturonan-binding domain [9,10]. Generally, the typical WAK proteins can be distinguished by several sequence characteristics, including an N-terminal signal peptide, an extracellular cell wall galacturonan-binding domain (GUB), an epidermal growth factor domain (EGF), a calcium-binding EGF domain (EGF-Ca), a transmembrane region (TM), and an intracellular conserved Serine/Threonine protein kinase (STK) domain [9,10]. The WAK-like (WAKL) proteins usually contain the extracellular cell wall galacturonan-binding domain (GUB) and intracellular conserved Serine/Threonine protein kinase (STK) domain, while other domains vary among different WAKL proteins [9]. Several studies have demonstrated that some WAK or WAKL proteins play an important role in plant innate immunity to pathogens. The extracellular domains of WAKs or WAKL proteins are involved in cytoplasm–cell wall communication, while the intracellular kinase domain is related to the activation of cytoplasmic signaling cascades in plant defense responses [11]. When plants are invaded by pathogens, WAK or WAKL proteins can perceive short oligogalacturonic acid fragments (OGs) released by plant cell wall wounding, and trigger defense signaling in plants. In *Arabidopsis*, the AtWAK1 is significantly induced by OGs, initiating immunity against the necrotrophic fungus *Botrytis cinerea* [12]. *AtWAKL22* and *AtWAKL10* participate in resistance to *Fusarium* wilt disease [13] and defense against *Pseudomonas syringae*, respectively [14]. In maize (*Zea mays*), ZmWAK-RLK1 (encoded by *Htn1*) and *Zm*WAK *qHSR1* confer quantitative resistance to the northern corn leaf blight and maize smut diseases, respectively [15,16]. In rice (*Oryza sativa*), *OsWAK1*, *OsWAK14*, *OsWAK91*, and *OsWAK92* contribute to resistance against the blast fungus *Magnaporthe oryzae* [17,18], and the *WAK* gene *Xa4* could confer durable resistance against *Xanthomonas oryzae pv.oryzae* [19]. In wheat (*Triticum aestivum*), *Stb6*, encoding a WAKL protein, confers resistance against *Zymoseptoria tritici* [20], and *TaWAK6* participates in wheat adult plant resistance to leaf rust (*Puccinia triticina*) [21]. Our laboratory showed that two wheat WAK genes, *TaWAK2A-800* and *TaWAK-6D*, and a *TaWAKL* gene *TaWAK-7D* (lacking the signal peptide in protein) participate in defense responses to *Fusarium graminearum*, *R. cerealis*, and *F. pseudograminearum* [22,23,24]. In *Gossypium hirsutum*, the wall-associated kinase GhWAK7A positively regulates cotton response to infections of both soil-borne fungal pathogens *Verticillium dahliae* and *Fusarium oxysporum f.* sp. *vasinfectum* [25].

Based on a high-quality version of a plant genome assembled, genome-wide identification of the *WAK* gene family has been reported in several plant species, such as *Arabidopsis* [9], rice [26], tomato [27], cotton [28], barley [29], and wheat [30]. For instance, in the *Arabidopsis* genome, there are five *WAK* and twenty-one *WAKL* genes, and the extracellular domain of WAKL proteins only show 18% to 22% identities to the WAKs [9]. In rice genome, 125 genes were identified as belonging to the WAK subfamily [26]. In tomato, 11 typical *SlWAK* and 18 *SlWAKL* genes were identified from the tomato genome, and these *SlWAK* or *SlWAKL* genes were distributed in nine of twelve chromosomes [27]. In cotton, 29 *GhWAK* genes were identified, and all cotton GhWAK proteins can be divided into five clades [28]. In barley genome, 91 *HvWAK* genes were identified [29]. Recently, 320 genes belonging to the WAK subfamily containing WAK_GUB domain, TM domain, and protein kinase domain were identified from wheat genome using the Pfam and SMART websites with three domains. They were divided into three phylogenetic groups [30]. However, the *WAK* and *WAKL* genes in wheat were not distinguished according to the typical characteristics of WAK kinase.

In this study, we genome-widely identified the typical *WAK* (not *WAKL*) genes in the wheat genome. According to the conserved characteristics of *Arabidopsis* WAK1-5, we identified a total of 140 *TaWAK* candidate genes in the wheat genome. These *TaWAK* genes were located on the 20 chromosomes of wheat genome, except chromosome 4B. By analyzing the RNA-sequencing (RNA-seq) data of wheat against *R. cerealis* and *F. pseudograminearum*, we found that a *TaWAK* gene *TaWAK-5D600* (*TraesCS5D02G268600*) located on chromosomes 5D was significantly upregulated, and that its upregulated transcript levels in response to both pathogens were higher compared with other *TaWAK* genes. Knock-down of *TaWAK-5D600* impaired wheat resistance against the fungal pathogens *R. cerealis* and *F. pseudograminearum* in wheat and repressed expression of defense-related genes in wheat, *TaSERK1*, *TaMPK3*, *TaPR1*, *TaChitinase3*, and *TaChitinase4*. This study proposes *TaWAK-5D600* as a promising gene for enhancing broad resistance to sharp eyespot and *Fusarium* crown rot (FCR) in wheat.

## 2. Results

### 2.1. Identification of Typical WAK Family Genes in Wheat

WAK proteins were characterized by an N-terminal signal peptide, a galacturonan binding domain (GUB), an EGF-like domain, a calcium binding EGF domain (EGF-Ca), a transmembrane domain, and an intracellular C-terminal STK domain [9,25]. In this study, we used the five *Arabidopsis* AtWAK proteins as queries, and first performed a local BLAST search (*e*-value < 1 × 10^−10^) against the wheat genome database [31]. Secondly, the conserved domains of these candidate proteins were manually predicted through the SMART website (http://smart.embl-heidelberg.de/ accessed on 24 August 2022). Based on the conserved characteristics of *Arabidopsis* WAK1-5, proteins lacking typical domains of WAKs were screened out from the candidate proteins. As a result, a total of 140 *TaWAK* candidate genes, encoding typical WAK proteins possessing an N-terminal signal peptide, a GUB domain, an EGF-like domain, an EGF-Ca domain, a transmembrane domain, and an intracellular conserved STK domain, were identified from the wheat genome [1]. These *TaWAK* genes were located on 20 wheat chromosomes, but not on chromosome 4B (Figure 1). The majority of *TaWAK* genes are tandemly arranged on each chromosome, and the quantities of *TaWAK* genes from the most to the least are chromosomes 6 (45 *TaWAK* genes), chromosomes 5 (33 *TaWAK* genes), chromosomes 2 (25 *TaWAK* genes), chromosomes 1 (20 *TaWAK* genes), chromosomes 7 (nine *TaWAK* genes), chromosomes 3 (five *TaWAK* genes), and chromosomes 4 (three *TaWAK* genes). The length of these TaWAK proteins ranged from 663 to 791 (amino acid, aa), with an average of 719.84 aa. The molecular weight values of these TaWAKs proteins ranged from 71.19 kDa to 87.05 kDa, with an average of 79.40 KDa, and isoelectric point (pI) values ranged from 4.89 to 8.66, with an average of 6.57 (Appendix A).

### 2.2. TaWAK-5D600 Are Involved in Wheat Responses against R. cerealis and F. pseudograminearum

To investigate the potential roles of *TaWAKs* in defense responses against infection of *R. cerealis*, we first analyzed the RNA-seq data of wheat inoculated with *R. cerealis*. Moreover, to explore which *TaWAK* genes were involved in wheat resistance to *F. pseudograminearum*, we also checked the transcript levels of whole 140 *WAK* genes in the online RNA-seq data (http://www.wheat-expression.com/ accessed on 25 August 2022) [32]. The results indicated that, after inoculating with *R. cerealis* or *F. pseudograminearum*, the expression of *TaWAK* genes presented a similar tendency in wheat, and compared with the mock treatment, 36 of the 140 *TaWAK* genes were upregulated after inoculating with both of the two pathogens (Figure 2A,B). This suggested that some *TaWAK* genes involved in defense responses might be conserved between *R. cerealis* and *F. pseudograminearum*.

To further analyze the transcript profiles of these upregulated *TaWAK* genes, we calculated the transcript changing fold of these *TaWAK* genes upon the *R. cerealis.* As shown in Figure 3A, among the whole 140 *TaWAKs* in wheat, 27 *TaWAK* genes were upregulated higher than 1.5-fold at 4 d post-inoculation (dpi) with *R. cerealis* compared to the mock treatment. Additionally, we also calculated the transcript change fold of these 27 *TaWAK* genes upon the *F. pseudograminearum*. In total, 11 of the 27 *TaWAK* genes were also upregulated higher than 1.5-fold at 36 h inoculated with *F. pseudograminearum* compared to the mock treatment (Figure 3B). In particular, two homologous genes on chromosomes 5A and 5D, *TraesCS5A02G261200* (named *TaWAK-5A200*) and *TraesCS5D02G268600* (named *TaWAK-5D600*), showed higher transcript fold, and upregulated levels responding to both pathogens compared with other *TaWAK* genes. As shown in Figure 3A, the transcript levels of *TaWAK-5A200* and *TaWAK-5D600* were elevated to 4.17-fold and 4.33-fold at 4 dpi with *R. cerealis* compared with mock treatment and increased to 5.06-fold and 17.91-fold at 10 dpi with *R. cerealis* compared with mock treatment, respectively. Additionally, at 36 h inoculated with *F. pseudograminearum*, the transcript levels of *TaWAK-5A200* and *TaWAK-5D600* were elevated to 1.91-fold and 3.73-fold compared with mock treatment (Figure 3B). These data suggest that the two homoeologous *TaWAK* genes, especially *TaWAK-5D600*, might be involved in wheat resistant responses to both sharp eyespot and FCR. Here, we focused on the functional investigation of *TaWAK-5D600.*

Subsequently, RT-qPCR assays were used to verify the transcript patterns of *TaWAK-5D600* in wheat stems/sheaths infected in time-course by *R. cerealis* or *F. pseudograminearum*. As shown in Figure 3C, *TaWAK-5D600* transcript was significantly upregulated after *R. cerealis* infection and peaked at 10 dpi (~8.56-fold over non-treatment). Furthermore, *TaWAK-5D600* transcript was significantly upregulated toward *F. pseudograminearum*. As shown in Figure 3D, *TaWAK-5D600* transcript abundance increased to 1.68- and 3.54-fold at 1 dpi and 2 dpi with *F. pseudograminearum* compared with non-treatment. Taken together, the transcript patterns of *TaWAK-5D600* indicate that the gene was involved in wheat defense responses to both *R. cerealis* and *F. pseudograminearum* infections.

### 2.3. Phylogenetic and Sequence Analyses, and Sub-Cellular Localization of TaWAK-5D600

To deepen the understanding of *TaWAK-5D600*, we first conducted phylogenetic analyses of TaWAK-5D600 protein and some immunity-associated WAK proteins from different plant species. These WAKs are roughly grouped into five categories (Figure 4A). The first branch included *T. aestivum* TaWAK-5D600, TaWAK-7D [23], TaWAK-6D [24], and *Aegilops tauschii* AeWAK3. The second branch included *A. thaliana* AtWAK1 [12], AtWAK2 [33], and *G. hirsutum* GhWAK7A [25]. The third branch included *A. thaliana* AtWAKL10 [13], AtWAKL22 [14], *O. sativa* OsWAK91 [18], and OsWAK92 [18]. The fourth branch included *O. sativa* OsWAK1 [17], *T. aestivum* TaWAKL4 (Stb6) [20], and *Z. mays* ZmWAK-RLK1 (Htn1) [15]. The last branch included *O. sativa* OsWAK-Xa4 [19], *T. aestivum* TaWAK2A-800 [22], TaWAK6 [21], and *Z. mays* ZmWAK-qHSR1 [16]. Meanwhile, *O. sativa* OsWAK14 was not clustered with other WAK proteins. These results suggest that TaWAK-5D600 was highly similar to *A. tauschii* AeWAK3 (95.39% identity). TaWAK-5D600 was also evolutionarily close to sharp eyespot-resistant protein *T. aestivum* TaWAK-7D [23] and sharp eyespot- and *Fusarium* crown rot-resistant protein TaWAK-6D [24], implying TaWAK-5D600 might have a similar function to TaWAK-7D and TaWAK-6D.

Additionally, pairwise sequence alignment showed that the TaWAK-5D600 protein only shared 20.00%, 27.05%, 28.25%, 24.5%, 31.06%, 33.37%, 14.49%, 20.21%, 33.33%, 32.63%, 40.65%, and 49.68% proteins sequence identityies with previously reported resistance-related WAKs, *Z. mays* ZmWAK-RLK1 (*Htn1*), ZmWAK-qHSR1, *O. sativa* OsWAK-Xa4, OsWAK1, OsWAK91, OsWAK92, OsWAK14, *T. aestivum* TaWAKL4 (*Stb6*), TaWAK2A-800, TaWAK6, TaWAK-7D, and TaWAK-6D (Appendix A). These results suggest that TaWAK-5D600 is distinct from these reported resistance-associated WAK proteins in crop plants.

*TaWAK-5D600* sequence cloned from the resistant wheat cultivar CI12633 contains an open reading frame with 2232 bp length sequence (Figure 4B). The predicted TaWAK-5D600 protein consists of 743 amino acid (aa) residues with a predicted Mw of 82.02 kDa and theoretical pI of 6.15. As shown in Figure 4C, the TaWAK-5D600 protein contains a signal peptide (no. 1–28 aa), a cell wall extracellular GUB domain (no. 35–142 aa), an EGF domain (no. 248–296 aa), an EGF-Ca domain (no. 297–336 aa), a transmembrane region (no. 348–370 aa), and an intracellular STK domain (no. 420–689 aa).

To investigate the sub-localization of TaWAK-5D600 in wheat cells, the p35S:GFP-TaWAK-5D600 fusion vector was constructed, and then the resulting p35S:GFP-TaWAK-5D600 vector or control p35S:GFP vector DNAs were separately introduced into wheat mesophyll protoplasts and transiently expressed (Figure 4D). The confocal microscopic observation results showed that GFP-TaWAK-5D600 protein mainly distributed at the plasma membrane, while the control p35S:GFP protein diffused both in the nucleus and cytoplasm. The results indicate that TaWAK-5D600 mainly localizes at the plasma membrane.

### 2.4. TaWAK-5D600 Is Required for Wheat Defense against R. cerealis and F. pseudograminearum

To identify the defense function of *TaWAK-5D600* against *R. cerealis* and *F. pseudograminearum* in wheat, the *TaWAK-5D600* was silenced by a Barley stripe mosaic virus (BSMV) induced silencing (VIGS) assay. As shown in Figure 5A, at 15 dpi infected by BSMV, typical stripe virus infection symptoms appeared on the wheat leaves and the transcript of BSMV coat protein (*CP)* gene was detected, which indicates the BSMV had successfully infected the test wheat plants. In addition, RT-qPCR assays results showed the transcript abundance of *TaWAK-5D600* was significantly lower in the BSMV: TaWAK-5D600 plants than in BSMV:GFP (control) CI12633 plants, indicating the *TaWAK-5D600* was successfully silenced in the VIGS wheat plants (Figure 5B).

Subsequently, these *TaWAK-5D600*-silenced or BSMV:GFP (control) wheat seedlings were inoculated with *R. cerealis* strain Rc207 or *F. pseudograminearum* WHF220 as previously described [8,34]. At 30 dpi with *R. cerealis* Rc207, the *TaWAK-5D600*-silenced wheat seedlings displayed more serious disease symptoms of sharp eyespot, including bigger necrotic areas and higher infection types (ITs) (Figure 5C). As shown in Figure 5D and Appendix A, the disease severity quantified results showed that the average ITs of *TaWAK-5D600*-silenced wheat seedlings were 2.65 and 2.50, and the corresponding disease indexes (DIs) were 53 and 50, while the average ITs of BSMV: GFP (control) CI12633 seedlings were 1.40–1.45, and their disease indexes were 28 and 29 in two VIGS batches. These results indicate that *TaWAK-5D600*-silenced wheat plants were more susceptible than the control wheat plants, and the *TaWAK-5D600* was required for wheat resistance against *R. cerealis*.

In addition, after inoculation with *F. pseudograminearum* WHF220, the *TaWAK-5D600*-silenced wheat plants also exhibited greater disease severity of *Fusarium* crown rot symptoms than the BSMV:GFP-infected CI12633 plants (Figure 5E). Furthermore, the disease severity quantified results showed that the average disease indexes were 38.8 and 37.6 (corresponding ITs of control wheat plants were only 2.9 and 2.8), while the average disease indexes of *TaWAK-5D600*-silenced plants were 67.5 and 61.2 (corresponding ITs were 5.0 and 4.5), respectively (Figure 5F, Appendix A). These results suggest that *TaWAK-5D600* was also required for wheat resistance to *F. pseudograminearum*.

### 2.5. TaWAK-5D600 Positively Regulated Expression of Several Defense-Related Genes

The RT-qPCR assays were used to examine transcription profiles of wheat defense-related genes in *TaWAK-5D600*-silenced and BSMV:GFP (control) wheat seedlings. As shown in Figure 6, the transcript levels of *TaSERK1*, *TaMPK3*, *TaPR1*, *TaChitinase3*, and *TaChitinase4* were significantly lower in *TaWAK-5D600*-silenced CI12633 wheat seedlings compared with the BSMV:GFP (control) wheat seedlings, suggesting that *TaWAK-5D600* regulates positively the expression of several defense-associated genes in wheat seedlings.

## 3. Discussion

Genome-wide sequences analysis of the *TaWAK* genes could provide valuable information for identifying new resistant genes against fungal pathogens in wheat. Xia et al., using the *Arabidopsis* AtWAK/AtWAKL proteins from *Arabidopsis* as queries, filtered all candidate genes using three conserved domains of AtWAK/AtWAKL (WAK_GUB domain, TM domain, and protein kinase domain), eventually identifying 320 *TaWAK* genes in wheat genome [30]. However, in their study, the *WAK* and *WAKL* genes are not strictly distinguished according to the typical characteristics of WAK kinase. In this study, we mainly focused on the typical *WAK* (not *WAKL*) genes in the wheat genome. Using *Arabidopsis* WAK1-5 as queries, according to the conserved characteristics of WAK proteins (a WAK_GUB domain, an EGF-like domain, an EGF-Ca domain, a transmembrane domain (TM), and an intracellular cytosolic STK domain) [9,25], a total of 140 *TaWAK* candidate genes were identified from the wheat genome. This work will be helpful to better understand the roles of *TaWAK* gene members in wheat.

To further investigate the potential functions of *TaWAK* genes in defense responses against the two pathogens, we checked the transcript levels of whole 140 *TaWAK* genes in wheat RNA-seq data upon *R. cerealis* infection, and the online RNA-seq data upon *F. pseudograminearum* infection (http://www.wheat-expression.com/ accessed on 25 August 2022) [32]. The results indicated that with *R. cerealis* or *F. pseudograminearum* infection, the expression of *TaWAK* genes presented a similar tendency in wheat. It suggested that some *TaWAK* genes involved in defense responses were conserved between *R. cerealis* and *F. pseudograminearum*, and 11 *TaWAK* genes were significantly upregulated higher than 1.5-fold compared to the mock treatment both upon the *R. cerealis* and *F. pseudograminearum* infections. Among them, the two homologous genes (*TraesCS5A02G261200*, *TraesCS5D02G268600*), especially the *TaWAK* on chromosome 5D, showed higher transcript fold change, and higher transcript expression levels response to both pathogens compared with other *TaWAK* genes. Furthermore, the RT-qPCR assays also showed that the *TaWAK-5D600* gene was involved in wheat defense responses both against *R. cerealis* and *F. pseudograminearum*, and it was therefore selected for further experimental analysis by the barley yellow dwarf virus (BSMV)-mediated gene silencing (VIGS) assays. Silencing of *TaWAK-5D600* impaired wheat resistance to both of the two pathogens in wheat. Thus, our data revealed that *TaWAK-5D600* was a promising gene for enhancing wheat broad resistance to both sharp eyespot and FCR. Besides resistance in planted wheat cultivars, the epidemics and severity of both sharp eyespot and FCR diseases are also influenced by temperature and humidity in diverse agro-ecological conditions. In this study, we mainly focused on the transcript changing fold of *TaWAK* genes upon the infection of the *R. cerealis* and *F. pseudograminearum* and validated the resistance function of *TaWAK-5D600* to both sharp eyespot and FCR by VIGS assays. Additionally, by aligning the sequences of *TaWAK-5D600* and allele genes in natural populations, we found that there are many variation sites in the genome sequence of *TaWAK-5D600* among natural populations (Appendix A, http://wheat.cau.edu.cn/WheatUnion accessed on 28 February 2023) [35], suggesting that the resistance function of *TaWAK-5D600* might be differentiated among various wheat varieties. In the future, the gene may be applied in breeding resistant wheat varieties by molecular marker-assistant selection and gene-editing methods.

The phylogenetic analysis results showed that the TaWAK-5D600 were clustered on the same branch with *T. aestivum* sharp eyespot and FCR-resistant protein TaWAK-6D [24] and sharp eyespot-resistant protein TaWAK-7D [23]. Meanwhile, pairwise sequence alignment showed that the TaWAK-5D600 protein only shared 40.65% and 49.68% proteins sequence identities with TaWAK-7D and TaWAK-6D, suggesting that TaWAK-5D600 is a novel WAK protein involved in wheat disease resistance responses. The current subcellular-localization analysis showed that GFP-TaWAK-5D600 protein mainly distributed at the plasma membrane in wheat mesophyll protoplasts. This was consistent with the sub-localization of some WAKs proteins reported [23,24,25,36,37]. Many plasma membrane-localized RLK proteins were shown to be responsible for triggering a series of intracellular innate immune responses during pathogens infection. For instance, in cotton, silencing of *GhWAK7A* repressed the expression level of *GhMPK3* and *GhWRKY30* relative to control plants [25]. In rice, *OsWAK25* overexpression increased the expression level of *PR10* and *PBZ1* [37]. The maize smut-resistant gene *ZmqHSR1* could also elevate the transcript level of *ZmPR-1* and *ZmPR5* during pathogen infection [16]. In previous studies, a series of defense-related genes, such as *TaSERK1*, *TaMPK3*, *TaPR1*, *TaChitinase 3*, and *TaChitinase 4*, have been proven to be related to innate immunity of wheat to *R. cerealis* [38,39]. In this study, silencing of *TaWAK-5D600* not only repressed the transcript levels of immunity-related genes (*TaSERK1*, *TaMPK3*, *TaPR1*, *TaChitinase 3*, and *TaChitinase 4*) in wheat, but also compromised wheat resistance to both sharp eyespot and FCR infection. These data suggest that *TaWAK-5D600* positively participates in wheat innate immunity against the two fungal pathogens.

## 4. Materials and Methods

### 4.1. Plants and Fungal Materials

A sharp eyespot- and *Fusarium* crown rot-medium-resistant wheat cultivar, CI12633, was used in the BSMV-mediated silencing assays. All wheat seedlings were cultured for 14 h under 23 °C light conditions and 10 h under 15 °C dark conditions in greenhouse. The sharp eyespot pathogen (*Rhizoctonia cerealis* strain Rc207) and *Fusarium* crown rot pathogen (*Fusarium pseudograminearum strain* WHF220) were isolated and donated by Prof. Jinfeng Yu and Dr. Li Zhang (Shandong Agricultural University, China). The sharp eyespot-resistant wheat RILs used in RNA-sequencing were kindly donated by Prof. Jizeng Jia (Chinese Academy of Agricultural Sciences).

### 4.2. Identification of WAK Gene Family Members in Wheat

The *TaWAK* genes were identified according to previously published methods [9,25]. At first, the protein sequences of AtWAK1-5 were used as queries to perform local BLAST searches (*e*-value < 1 × 10^−10^) against the wheat genome database by utilizing the software Tbtools [1,31]. Subsequently, the conserved domains of the remaining candidate protein sequences were manually checked using the online SMART website (http://smart.embl-heidelberg.de/ accessed on 24 August 2022), and proteins with typical characteristics of TaWAK proteins were screened out. Eventually, proteins with an N-terminal signal peptide, an extracellular cell wall galacturonan-binding domain (GUB), an epidermal growth factor domain (EGF), a calcium-binding EGF domain (EGF-Ca), a transmembrane region (TM), and an intracellular conserved Serine/Threonine protein kinase (STK) domain were regarded as TaWAK family members in wheat.

### 4.3. Phylogenetic, Gene Structure, and Conserved Motifs Analysis

The neighbor-joining method (MEGA 6.0) was used to construct the phylogenetic tree through the neighbor-joining method with 1000 bootstrap replicates. The basic characteristics of protein sequences were predicted through the online software ExPASy7 (https://www.expasy.org/ accessed on 27 August 2022).

### 4.4. RNA-Seq Data Analysis

At mock treatment and 4 and 10 dpi with *R. cerealis* infection, the leaf sheaths of the resistant recombinant inbred lines (RILs-R) wheat seedlings derived from the cross ‘Shanhongmai’ × ‘Wenmai 6′ were used for RNA extraction and RNA sequencing [22]. To explore which *TaWAK* genes were involved in wheat resistance to *R. cerealis*, we checked the transcript levels of whole 140 *TaWAK* genes in the RNA-seq data of *R. cerealis* infection.

### 4.5. RNA Extraction and RT-qPCR

At non-treatment, 4, 7, and 10 dpi with *R. cerealis* Rc207, as well as at non-treatment, 1, and 2 dpi with *F. pseudograminearum* infection, the purified RNA were extracted from CI12633 wheat seedlings using a Trizol reagent (Invitrogen, Waltham, MA, USA). In VIGS assays, after 15 dpi with BAMV infection, the purified RNA were extracted from BSMV: GFP (control) and BSMV:TaWAK-5D600-infected CI12633 wheat seedlings. Then, the RNA was reverse-transcribed into cDNA with a FastQuant RT Kit (Tiangen, Beijing, China). Eventually, specific primers were used to check the transcript level of *TaWAK-5D600*, as well as several defense-related genes *TaSERK1*, *TaMPK3*, *TaPR1*, *TaChitinase3*, and *TaChitinase4* (Appendix A) [40].

### 4.6. Subcellular Localization of TaWAK-5D600

Using the specific primers TaWAK-5D600-GFP-inF1/R1, the coding region sequences of *TaWAK-5D600* were amplified from the cDNA of CI12633 wheat seedlings. Then, the coding region sequences of *TaWAK-5D600* were sub-cloned into the 5′-terminus of the green fluorescent protein (GFP), forming a pCaMV35S: TaWAK-5D600-GFP fusion construct. Subsequently, the p35S: GFP empty construct or p35S: TaWAK-5D600-GFP fusion construct was introduced into wheat protoplasts, respectively [41]. After darkness incubation at 25 °C for 16 h, the confocal laser scanning microscope (ZeissLSM 700, Germany Carl Zeiss, Oberkochen, Germany) with a Fluor × 10/0.5M27 objective lens and SP640 filter was used to observe the GFP signals.

### 4.7. VIGS and Assessment for Wheat Resistance to R. cerealis and F. pseudograminearum

The Barley stripe mosaic virus (BSMV)-mediated silencing (VIGS) assays were used to assess the resistant function of *TaWAK-5D600* against *R. cerealis* or *F. pseudograminearum*. A 192 bp fragment of *TaWAK-5D600* in the anti-sense orientation was sub-cloned into the RNA γ of BSMV, to form a BSMV: TaWAK-5D600 recombinant γ construct (Appendix A). Subsequently, the BSMV: TaWAK-5D600 and BSMV: GFP virus construct were transcribed into RNAs in vitro and used to infect CI12633 wheat seedlings at the three-leaf stage. After 15 days infected with BSMV, the fourth leaves of the BSMV-infected wheat seedlings were sampled to evaluate the silencing efficiency of *TaWAK-5D600*. Eventually, the *TaWAK-5D600*-silenced and the BSMV: GFP-infected CI12633 wheat seedlings were further used in resistance assessment to sharp eyespot or FCR as previously described [8,34].

## 5. Conclusions

We identified a novel cell wall-associated kinase gene, *TaWAK-5D600*, in wheat defense responses to both *R. cerealis* and *F. pseudograminearum* infection. *TaWAK-5D600* is required for wheat defense responses against the two pathogens and *TaWAK-5D600* positively regulates the expression of several immunity-related genes, including *TaSERK1*, *TaMPK3*, *TaPR1*, *TaChitinase 3*, and *TaChitinase 4*. Thus, *TaWAK-5D600* positively participates in defense against *R. cerealis* and *F. pseudograminearum* infections through modulating the transcript of several defense-related genes in wheat. *TaWAK-5D600* is a promising gene for enhancing wheat broad resistance to both sharp eyespot and FCR.

## Figures and Tables

**Figure 1 ijms-24-05060-f001:**
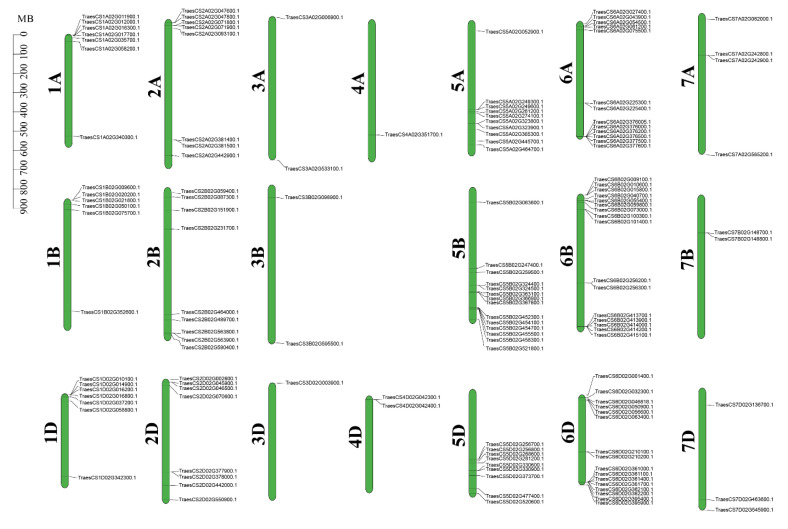
Chromosomal distribution of the typical *TaWAK* genes in wheat. The 140 *TaWAK* genes were unevenly distributed on 20 wheat chromosomes. The bar indicates the length of chromosome in megabases (MB).

**Figure 2 ijms-24-05060-f002:**
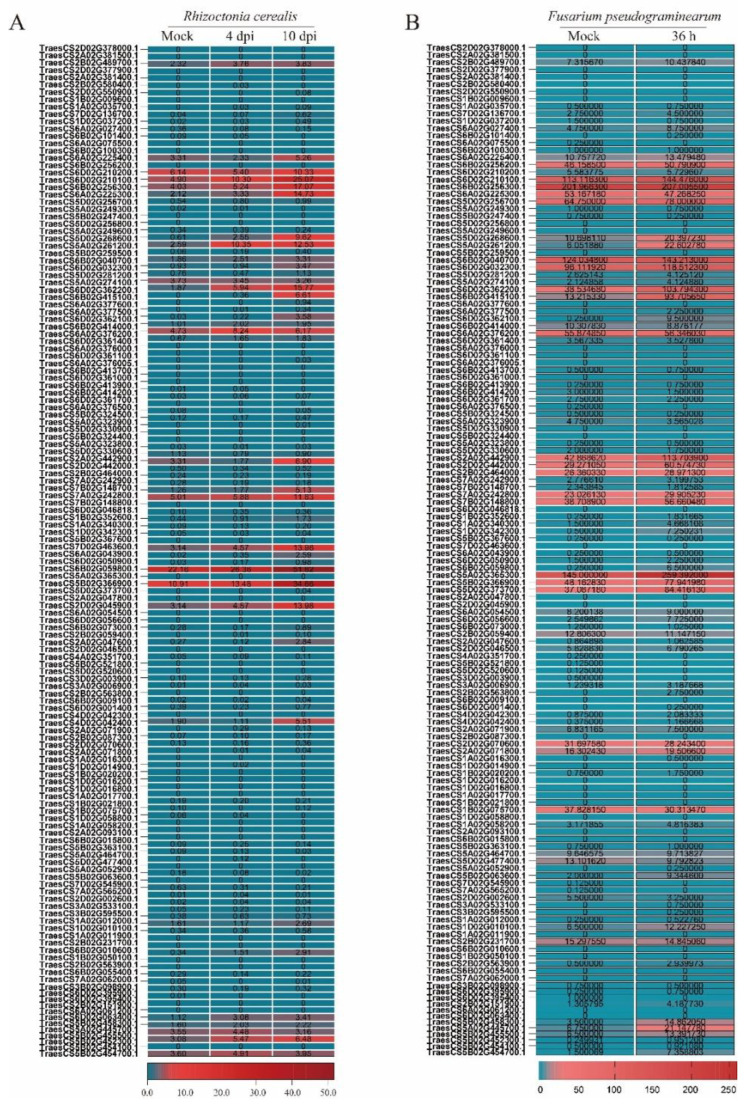
The transcript profiles of 140 *TaWAKs* in the wheat RNA-sequencing (RNA-seq) data. (**A**) The transcript levels of 140 *TaWAKs* upon *R. cerealis* infection in the resistant recombinant inbred lines (RILs) derived from the cross ‘Shanhongmai’ × ‘Wenmai 6′. (**B**) The transcript profiles of 140 *TaWAKs* upon *F. pseudograminearum* infection. The RNA-seq data upon *F. pseudograminearum* infection were checked from the online RNA-seq data (http://www.wheat-expression.com/ accessed on 25 August 2022) [32].

**Figure 3 ijms-24-05060-f003:**
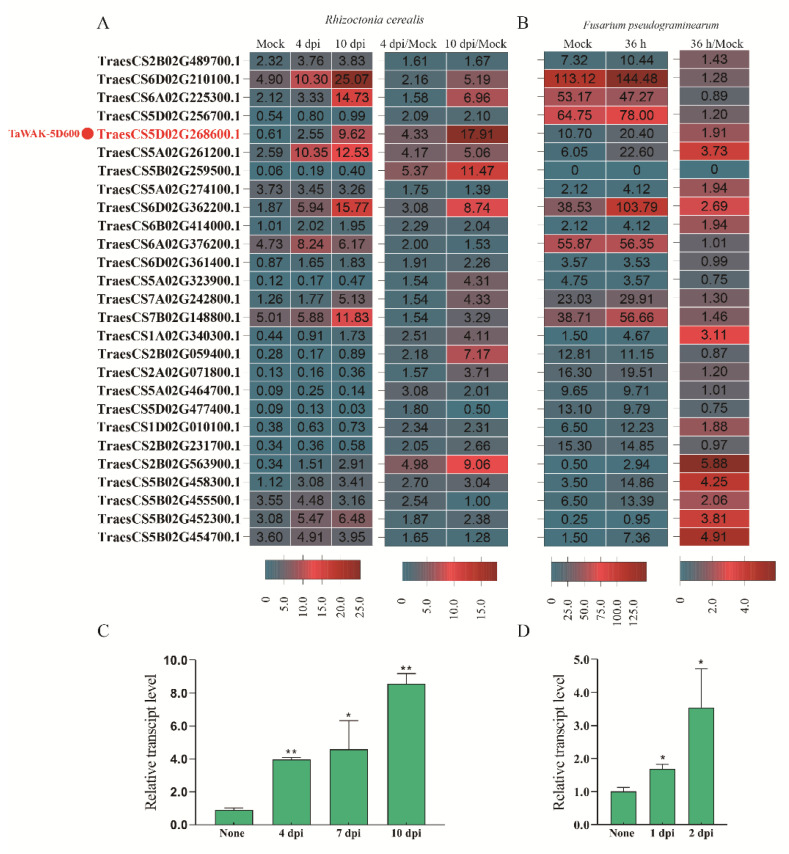
*TaWAK-5D600* is involved in wheat responses to both sharp eyespot and *Fusarium* crown rot. (**A**) The transcript levels and fold change of the significantly upregulated 27 *TaWAK* genes in the RILs-R response to *R. cerealis* infection. (**B**) The transcript levels and fold change of the 27 *R. cerealis* induced *TaWAK* genes upon *F. pseudograminearum* infection in the online RNA-seq data (http://www.wheat-expression.com/ accessed on 25 August 2022) [32]. (**C**) Transcript levels of *TaWAK-5D600* in sharp eyespot-resistant wheat line CI12633 at non-treatment and 4, 7, and 10 dpi with *R. cerealis* Rc207. (**D**) The transcript patterns of *TaWAK-5D600* in FCR-mildly-resistant wheat line CI12633 at non-treatment and 1 and 2 dpi with *F. pseudograminearum* WHF220. *TaWAK-5D600* transcript level at non-treatment is set to 1. *TaActin* gene was used as the internal control (*t*-test: * *p* < 0.05; ** *p* < 0.01).

**Figure 4 ijms-24-05060-f004:**
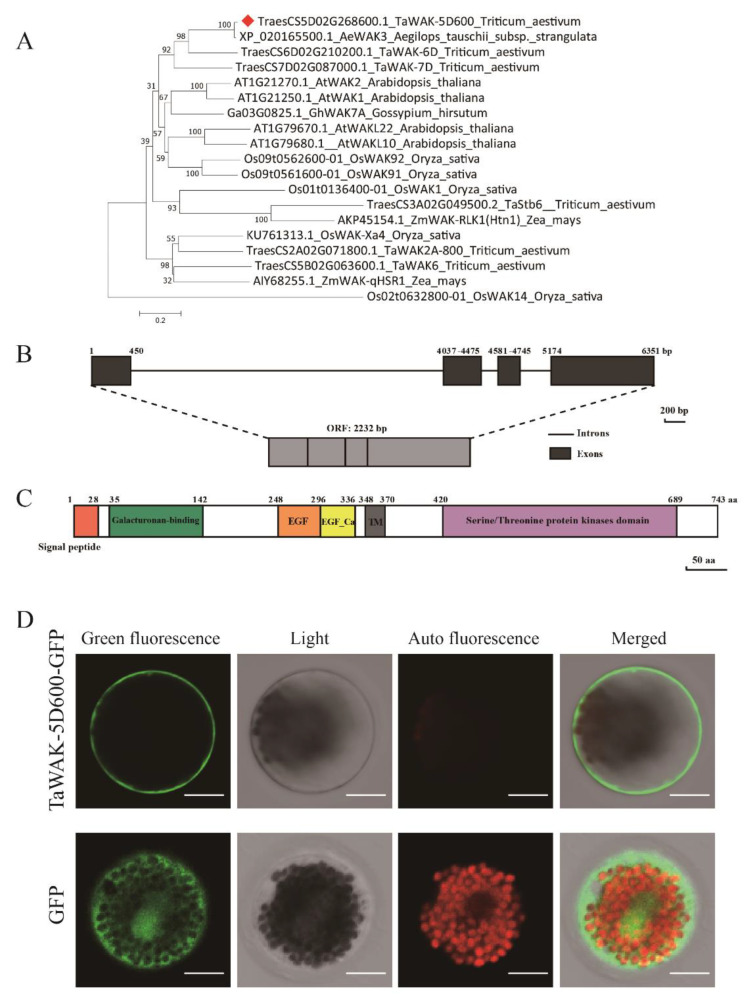
Phylogenetic tree, conserved-domain, and gene-structure analyses of *TaWAK-5D600*. (**A**) A phylogenetic tree of TaWAK-5D600 and other 18 WAK proteins from different plants. The position of TaWAK-5D600 was indicated by a red blot. (**B**) Gene structure of *TaWAK-5D600*; black boxes represent exons and black lines indicate introns. (**C**) Schematic diagram of the TaWAK-5D600 protein. The conserved protein domains of TaWAK-5D600 were represented by different colored boxes. (**D**) Subcellular localization of TaWAK-5D600 in wheat protoplasts cells (bars = 20 μm).

**Figure 5 ijms-24-05060-f005:**
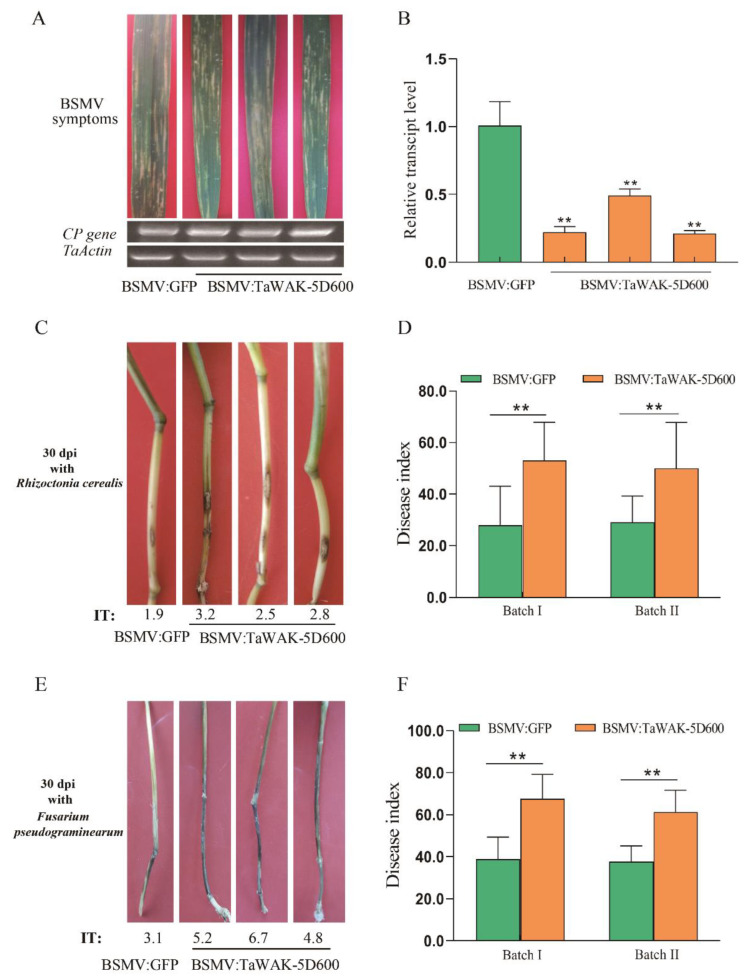
Silencing of *TaWAK-5D600*-compromised wheat resistance both to sharp eyespot and *Fusarium* crown rot. (**A**) Typical BSMV symptoms on wheat leaves at 15 dpi infected with BSMV and transcripts of BSMV coat protein (*CP*) gene detecting by RT-PCR assays. (**B**) The silencing efficiency of *TaWAK-5D600* detecting by RT-qPCR assay. The transcript level of *TaWAK-5D600* in BSMV:GFP (control) wheat seedlings was set to 1. (**C**) Sharp eyespot symptoms on *TaWAK-5D600*-silenced and BSMV:GFP (control) wheat plants at 30 dpi with *R. cerealis*. (**D**) Disease indexes (DIs) of *TaWAK-5D600*-silenced and BSMV:GFP (control) wheat plants at 30 dpi with *R. cerealis* in two independent batches (*t*-test: ** *p* < 0.01). (**E**) *Fusarium* crown rot symptoms on *TaWAK-5D600*-silenced and control wheat plants at 30 dpi with *F. pseudograminearum.* (**F**) Disease index (DI) of *TaWAK-5D600*-silenced and control wheat plants at 30 dpi with *F. pseudograminearum* WHF220 in two independent batches (*t*-test: ** *p* < 0.01). Bars indicate SEs of the mean.

**Figure 6 ijms-24-05060-f006:**
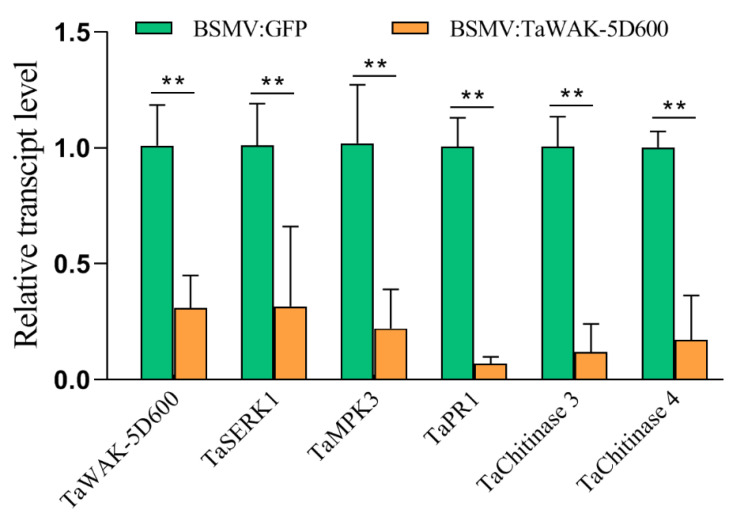
Transcript profiles of *TaWAK-5D600* and defense-related genes in BSMV:GFP (control) and BSMV:TaWAK-5D600-infected wheat seedlings. Relative transcript abundances of *TaWAK-5D600* and the tested genes *TaSERK1*, *TaMPK3*, *TaPR1*, *TaChitinase3*, and *TaChitinase4* in BSMV: TaWAK5D600-infected CI12633 seedlings were quantified relative to those in BSMV:GFP (control) seedlings. Statistically significant differences were calculated based on three replications via a Student’s *t*-test (** *p* < 0.01). *TaActin* was used as an internal control.

## Data Availability

All data supporting the findings of this study as well as Appendix A are available within the paper and published online.

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
