# Peer review of "A Novel Wall-Associated Kinase TaWAK-5D600 Positively Participates in Defense against Sharp Eyespot and *Fusarium* Crown Rot in Wheat"

_ijms, 2023, doi:10.3390/ijms24055060_

Round 1
Reviewer 1 Report
In this study, Haijun Qi and authors indicated that TaWAK-5D600 positively participates in defense against sharp eyespot and Fusarium crown rot in wheat. In general, the data analysis is in accordance with the conclusions, and the writing was easy to follow. I only want to suggest that one more paragraph on application of the results in the practice would be appreciated. For example, what environmental factors could affect described mechanisms, and how results of the study could contribute in resolving problems in control of F. pseudograminearum in diverse environmental conditions? Although the focus of the study is very clear and methodology is consistent with the results, I think that giving more information on how obtained results could be affected by environmental factors in diverse agro-ecological conditions could initiate more investigations with a multidisciplinary approach. Overall, I would recommend manuscript acceptance.
Author Response
Comments:
In this study, Haijun Qi and authors indicated that TaWAK-5D600 positively participates in defense against sharp eyespot and Fusarium crown rot in wheat. In general, the data analysis is in accordance with the conclusions, and the writing was easy to follow. I only want to suggest that one more paragraph on application of the results in the practice would be appreciated. For example, what environmental factors could affect described mechanisms, and how results of the study could contribute in resolving problems in control of F. pseudograminearum in diverse environmental conditions? Although the focus of the study is very clear and methodology is consistent with the results, I think that giving more information on how obtained results could be affected by environmental factors in diverse agro-ecological conditions could initiate more investigations with a multidisciplinary approach. Overall, I would recommend manuscript acceptance.
Reply: Dear reviewer: Sincerely thanks, we have added one paragraph in the discussion section in line 326-337 to discuss how to use the resistant gene TaWAK-5D600 in the future.
Reviewer 2 Report
After careful evaluation of the manuscript title “A novel wall-associated kinase TaWAK-5D600 positively par- 2 ticipates in defense against Sharp Eyespot and Fusarium Crown 3 Rot in Wheat”, you can find below my comments.
- The manuscript is understandable, correct and appropriate for International Journal of Molecular Sciences;
- The purpose of the manuscript and the problems to be solved at work are clearly defined, the purpose of the work is within the scope of International Journal of Molecular Sciences;
- The results are interesting and important for researchers from many research centers in the relevant fields of agricultural sciences, because they concern the health of one of the most important cereal plants, Triticum aestivum;
- Numbers are clear, correct and informative,, although figures 1-2 should be made clearer;
- The manuscript should be cited in the future and should generate great interest from readers in the area where the research was conducted.
I would like to suggest some corrections:
Results
Figure 1, 2 - please increase readability (if possible)
Discussion
Line 340 – are [23, 24, 25, 35, 36] should be [23-25, 35, 36]
Yours sincerely
Author Response
Comments: Results Figure 1, 2 - please increase readability (if possible).
Reply: Dear reviewer, sincerely thanks, we have added some descriptions of Figure 1 and Figure 2 in line 120-124 and 141-142 according to your comments.
Comments: Discussion Line 340 – are [23, 24, 25, 35, 36] should be [23-25, 35, 36]
Reply: Dear reviewer, sincerely thanks, we have revised the reference number in line 346.